# Supporting Immunization Uptake during a Pandemic, Using Remote Phone Call Intervention among Babies Discharged from a Special Neonatal Care Unit (SNCU) in South India

**DOI:** 10.3390/vaccines10040507

**Published:** 2022-03-25

**Authors:** Seema Murthy, Meenal Sawant, Sahana Sadholalu Doreswamy, Sateesh Chandra Pothula, Shirley Du Yan, Tanmay Singh Pathani, Deepali Thakur, Srikrishna Rajarama Sastry Vemuri, Sanjeev Upadhyaya, Shahed Alam, Madireddy Alimelu, Himabindu Singh

**Affiliations:** 1Aurora Health Innovations, Bengaluru 560038, India; seema@noorahealth.org (S.M.); sahana@noorahealth.org (S.S.D.); sateesh@noorahealth.org (S.C.P.); tanmay@noorahealth.org (T.S.P.); deepali@noorahealth.org (D.T.); 2Noora Health, Bengaluru 560038, India; shirley@noorahealth.org (S.D.Y.); shahed@noorahealth.org (S.A.); 3United Nations Children’s Fund India, Hyderabad 500034, India; rsvsrikrishna@gmail.com (S.R.S.V.); supadhyaya@unicef.org (S.U.); 4Niloufer Hospital, Hyderabad 500004, India; nilouferneonatology@gmail.com; 5Department of Neonatology and Pediatrics, Osmania Medical College, Hyderabad 500095, India; dr.himabindusingh@gmail.com

**Keywords:** SNCU babies, COVID, immunization uptake

## Abstract

COVID-19 has impacted children’s immunization rates, putting the lives of children at risk. The present study assesses the impact of phone-call counseling, on immunization uptake during the pandemic. Families of babies discharged from the SNCUs in six government centers in three South Indian states were recruited. Calls were made 10 days after the immunization due date. Missed vaccinees were counseled and followed up on 7 and 15 days. Of 2313 contacted, 2097 completed the survey. Respondents were mostly mothers (88.2%), poor (67.5%), and had secondary level education (37.4%). Vaccinations were missed due to the baby’s poor health (64.1%), COVID-19 related concerns (32.6%), and lack of awareness (16.8%). At the end of the intervention, the immunization uptake increased from 65.2% to 88.2%. Phone-call intervention can safely support immunization and lower the burden on health workers.

## 1. Introduction

Vaccine-preventable diseases (VPDs) such as rubella, measles, polio, and diphtheria, contribute significantly to child mortality and morbidity. Globally, they account for approximately 1.5 million deaths and 19.5 million serious morbidities among infants annually [1]. Full immunization can be a key, easy and cost-effective way to protect children from VPDs during the first 5 years of their life and improve child survival rates [2,3]. Globally, an estimated 2–3 million deaths could be prevented every year if every child completes the vaccination schedule [4]. However, most low-to-middle income countries (LMICs) such as Mediterranean, South-East Asia, and African regions, have less immunization coverage (80%) compared to high-income WHO regions (90%) [5].

Like most LMICs, India has been struggling to meet its immunization coverage goals. India has the largest birth cohort in the world and contributes significantly to child mortality and morbidity resulting from VPDs [6]. India has shown approximately a 19-percentage point increase in the national average of full immunization (BCG, Measles, and three doses each of Polio and DPT) since the NFHS-4 (2005-06). This might be one of the reasons for the recent decline in child mortality rates. However, India has a long way to reach its Sustainable Development Goal (SDG) [7]. Thus, increasing universal vaccination coverage continues to be a key strategy to achieve this goal. Additionally, in recent years, increasing resistance has been observed among parents to vaccinate their children between ages 12 and 23 years. Negative attitudes and beliefs about vaccines, lack of adequate information, distrust in the sources of information, and negative propaganda in social media (i.e., WhatsApp and Facebook) are believed to be primary reasons behind their hesitation [8,9].

The COVID-19 pandemic is expected to add to these existing challenges around maternal and child health care [10]. In April 2020, routine postnatal check-ups and immunization visits reportedly decreased compared to the previous year, thus putting mothers and children at higher risk of contracting VPDs [10,11,12]. In late March 2020, WHO recommended several LMICs to temporarily suspend routine immunization services and divert all health care resources towards controlling the COVID-19 pandemic [11]. In addition to the temporary hold on primary health services put by the Government of India, the shortage of frontline workers, frequent lockdowns, and lack of transportation during the pandemic period could have slowed down the utilization of most preventive services [10]. Simultaneously, fear of COVID-19 infection, lack of social distancing, and inadequate infection control practices affected health-seeking behaviour in general [10]. In light of all these challenges, there were concerns raised regarding relapse in immunization rates for India, Pakistan, and Nepal [12]. This highlights the need for appropriate efforts to continue immunization uptake in LIMCs [13]. Sick Neonatal Care Units’ (SNCUs) babies, due to their comorbidities, are prone to serious infections, making full immunization and adequate postnatal care even more important. Encouraging caregivers to immunize these babies in critical circumstances such as the COVID-19 pandemic is important.

The primary goal of the present study was to examine the impact of a phone-based intervention on immunization uptake in SNCU babies. In addition, we also assessed the extent of existing immunization coverage and reasons for not vaccinating the child.

## 2. Methods

### 2.1. Study Sites

The study was conducted in six health care facilities—(1) Niloufer Hospital, Hyderabad in the state of Telangana; (2) Rajiv Gandhi Institute of Medical Sciences (RIMS), Kadapa; (3) Maharani Hospital, Vizianagaram; and (4) Visakhapatnam Institute of Medical Sciences and Hospital (VMH), Visakhapatnam in the state of Andhra Pradesh; (5) Raichur Institute of Medical Sciences (RIMS), Raichur; and (6) Yadgir District Hospital in the State of Karnataka. Surveys were conducted from June to September 2020.

### 2.2. Study Participants

As shown in Table 1, a list of families (*n* = 3115) who had a baby discharged from the SNCU wards was obtained from respective facilities. Of these 3115, 802 (25.7%) families were excluded due to reasons such as the baby or mother’s death before or after discharge or they were unreachable by phone (i.e., switched off phones or invalid numbers). The remaining 2313 families who were eligible to participate were those who had reported the baby and mother alive during the time of call, spoke at least one of three languages (Hindi, Kannada, and Telugu), and had a valid phone number. From these 2313 families, 2097 (90.6%) families successfully completed the survey.

### 2.3. Study Procedures

Face validity of the structured questionnaire was established by experts and was pilot-tested with ten respondents. After taking oral consent from each respondent, tele-trainers collected information on the respondent’s socio-demographic background, baby’s health (i.e., age, last recorded weight, and immunization status), and detailed reasons for non-vaccination through a phone call. The initial call was made seven to ten days after the due date for immunization. Survey CTO software version 2.70 was used to record responses [14]. At the end of each call, families who had not vaccinated their baby or had missed the scheduled vaccine were counseled for immunization. During counseling, tele-trainers emphasized the role of vaccines in the baby’s health; addressed any fears, myths, or doubts the family had; gave information about where they can vaccinate their baby; and, if asked, gave the contact details of the local ASHA worker. Families who had not vaccinated their baby or were unreachable during the initial call were followed-up on after seven days. If families did not vaccinate their baby at the first follow-up call, they were further counseled. The second follow-up call was conducted fifteen days after the initial call (Figure 1). Immunization status self-reported by the participant was recorded at each time point. Ethical approval for the study was obtained from the ACE Independent Ethics Committee.

## 3. Findings

### 3.1. Sociodemographic Characteristics

Respondents were mostly mothers, the majority of whom owned a below poverty level (BPL) card and had no education or less than secondary level education. Most families had received a vaccination card that contained information regarding routine immunization schedules for the baby. A small percentage of families had received the card but did not have it with them at the time of the survey (Table 2).

### 3.2. Immunization Coverage

During the initial call, 1368 (65.2%) babies were fully immunized (i.e., had not missed any scheduled vaccines), whereas 716 babies (34.1%) had missed at least one of the scheduled vaccines. Families of these 716 babies were counseled for immunization. During the first follow-up call, tele-trainers found that 301 (42%) of these 716 families vaccinated their baby, 292 (42.0%) had not vaccinated their baby, 5 babies had died, and 118 (16.5%) families could not be reached. These 410 (292 + 118) families were called after seven days from the first follow-up. During the second follow-up, an additional 181 (44.1%) babies were vaccinated, 160 (39.0%) were not vaccinated, 65 (15.8%) could not be reached and four (1.0%) babies had died (Figure 1).

Overall, the baby’s health status, probable COVID-19 pandemic effect, and lack of awareness or support (Figure 2) were reported as primary reasons for non-vaccination. Detailed responses at each timepoint are specified in Table 3.

### 3.3. Post Call Change in Immunization Uptake

At the first follow-up call, the immunization rate changed from 65.2% to 79.5% (22% increase). At the second follow-up call, the immunization rate changed from 79.5% to 88.2% (11% increase). Overall, at the end of 2–3 weeks a total of 1850 babies (i.e., 482 additional) were vaccinated and a 35% increase was seen in immunization uptake reducing the immunization gap in this sample (Figure 3).

We did not intend to measure the cost incurred during the study. However, post-study calculations showed an approximate expenditure of EUR 2 per conversion from non-vaccinated to vaccinated status. This included the cost of resources for training, training time, running phone costs, and time of data enumerators for making calls. It does not include other administrative costs and the time of experts for planning and executing the study.

## 4. Discussion

Child mortality due to inadequate or non-immunization is a serious public health issue in India. Every year, thousands of children die and several hundred are left at risk of developing serious disabilities because they are either partially or not vaccinated [15]. In India, although rates of full immunization have increased, there are geographical differences, with rural and low resource areas showing lower rates [16,17].

Interruption or temporary suspension of child immunization during the COVID-19 pandemic is expected to reduce the ability of healthcare systems to meet the need for routine maternal and child healthcare services [13]. Studies conducted around the 2014 Ebola outbreak show that disruption of the routine childcare services during the period had resulted in a second public health crisis in Africa [18]. In some communities of Sierra Leone, the age-eligible measles vaccination rate was reduced by 25.9 percentage points from 71.3% (before the Ebola outbreak) to 45.7% (during the outbreak) [19]. In our study population, conducted after the first wave of the COVID-19 pandemic, we found that 34.1% of families were not able to vaccinate their baby or had missed the vaccination schedule. COVID-19 pandemic challenges (such as lockdown, fear of infection, misinformation) were amongst the primary reasons for the delayed vaccination.

Studies evaluating the effect of programs to improve immunization uptake in LMICs are very limited. Programs that create awareness, provide health education to the community, and teach preventive behaviours are found to help increase knowledge and change attitudes and behaviours of the community [20,21,22,23]. Specifically, targeted interventions such as community or facility-based educational programs to caregivers, providing reminder cards to mothers, home visits by ASHAs or ANMs, and integrating immunization with other primary health care services have been found to be somewhat effective in increasing immunization uptake in LMICs [24]. For instance, after the Ebola outbreak, an 11-percentage point increase was found in measles vaccination rates after vaccinators visited home to educate parents and encourage them to vaccinate their babies [25]. A randomized controlled trial conducted in an urban area of Pakistan showed a 31% increase in the completion rate of the third dose of diphtheria–pertussis–tetanus (DPT3) among the intervention group (i.e., mothers who received a redesigned immunization card and canter-based education) compared with the standard of care group [26]. We did not find reports on phone-based interventions in crisis situations targeted at improving immunization.

The present study, conducted during the COVID-19 pandemic, includes a large sample of SNCU babies across three states in South India from six government facilities. We found that it was necessary to reach families and provide targeted information and counselling about vaccination to increase immunization during the COVID-19 pandemic. Considering the challenges the healthcare system faced during this period, a simple and remote intervention to counsel families was attempted. An increase in child immunization observed after two rounds of follow-up phone calls in our study needs further consideration and study to understand if it could be a viable additional support system.

The study has a few limitations. Selection bias could have occurred due to the fact that those people who had active phone connections and picked up calls were included. Therefore, the results of this study could only be applicable to similar populations. Immunization data were self-reported and did not corroborate hospital records. This might result in self-reporting bias. To minimize the impact of this bias on study results, we considered verifying vaccination status by asking participants to upload their vaccination cards. However, an initial exploration during the pilot study showed that challenges such as lack of smartphone availability, unfamiliarity with technology to upload photos, and quality of photos shared made it unfeasible. Furthermore, cross verifying with government records was also not possible given the issue of lack of access to personalized health records. We did not collect the vaccination date within the village and went by the birth date. However, we gave a gap of 10 days after the due date to ensure that this was not a limiting criterion. Data collection extended over a long period and continued after lockdowns were lifted. Thus, it is possible that factors such as the lifting of lockdowns, catch-up immunization drives might contribute to the intervention effect to some extent. However, since the majority of the data were collected after the lockdown was lifted and 33.3% of families reported COVID-19 and lockdown-related reasons for non-vaccination, we assume that this would have played a minor role. The results of this study are generalizable only to other high-risk babies with similar sociodemographic profiles and in similar health systems.

## 5. Conclusions

Upsurge in immunization uptake in this study revealed that persistent reminder phone calls that provide information and confidence to families could be a potentially viable, quick, safe, and convenient option to support the existing health system. When used in conjunction with existing interventions, these reminder calls can help connect with hard-to-reach populations and address their health-related needs in critical situations such as the COVID-19 pandemic. The results of our study are encouraging, but further studies are needed to address some of the limitations to build evidence. We believe that the learnings from this implementation research study may be applicable to other similar geographies and health systems, not only for this pandemic but other crisis situations as well.

## Figures and Tables

**Figure 1 vaccines-10-00507-f001:**
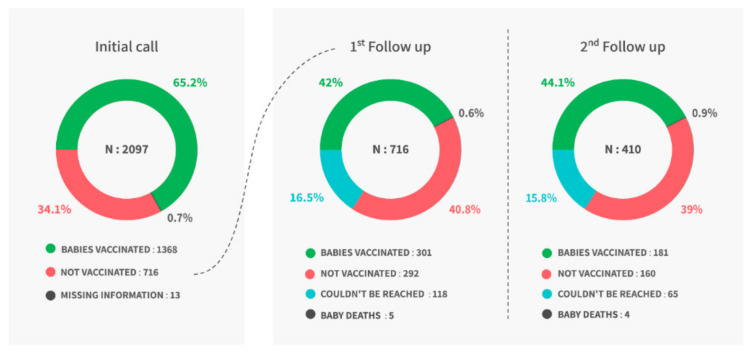
Follow-up schedule.

**Figure 2 vaccines-10-00507-f002:**
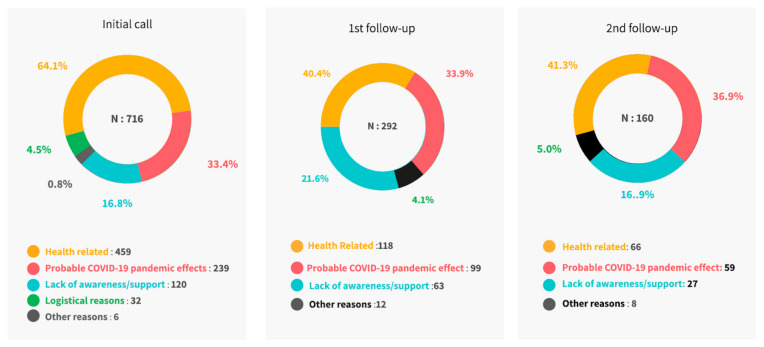
Overall reasons for non-vaccination.

**Figure 3 vaccines-10-00507-f003:**
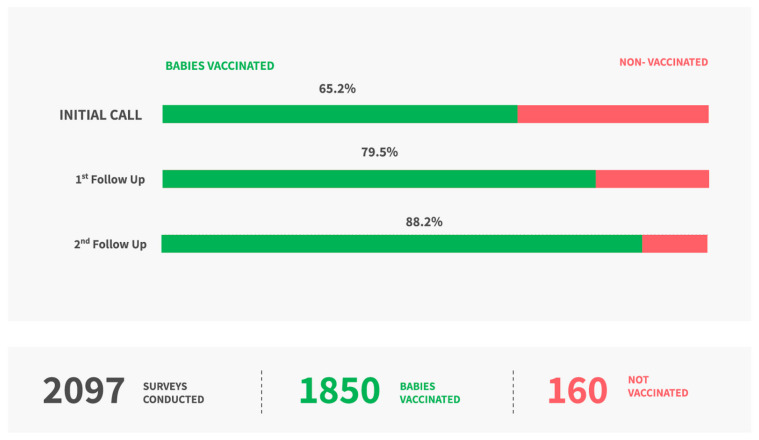
Immunization uptake at the end of the study.

**Table 1 vaccines-10-00507-t001:** Selection criteria for study participants.

Reason	Population Size (*N*)	Sample Size (*n*)	%
**List of Families Obtained from the Hospital**	**3115**		
** *Excluded from the study* **	3115	802	25.7%
Baby/mother died before discharge	802	173	21.5%
Baby/mother died after discharge	802	91	11.3%
Invalid/wrong number	802	197	24.5%
Switched off phones	802	332	41.4%
Data enumerator spoke a different language	802	9	1.1%
** *Included in the Study* **	*3115*	*2313*	*74.3%*
Included in the study and survey complete	2313	2097	90.6%
Included in the study but survey not complete	2313	216	9.3%
• Not available	216	71	32.8%
• Not picked up or answered the call	216	61	28.2%
• The family refused to participate	216	51	23.6%
• Baby not yet discharged	216	18	8.3%
• Other reasons *	216	15	6.9%

* Patient not responding, the baby born in another hospital, call disconnected or completed by another investigator.

**Table 2 vaccines-10-00507-t002:** Characteristics of study participants (*N* = 2097).

Variables	*n*	%
** *State-wise Respondents* **		
Andhra Pradesh	948	45.2%
Karnataka	732	34.9%
Telangana	417	19.9%
** *Respondents* **		
Mother	1849	88.2%
Father	128	6.1%
Other	120	5.7%
** *Education Level* **		
No education	399	19.0%
No formal education but can read and write	23	1.1%
Up to 5th standard	126	6.0%
6th to 10th standard	785	37.4%
11th standard to degree/diploma	575	27.4%
Graduate	137	6.5%
Post-graduate	52	2.5%
***Owns BPL*** * ***Card***	1415	67.5%
** *Received Vaccination Card* **	1996	95.2%
Received and accessible at home	1922	96.3%
Received but not accessible	74	3.7%
Did not receive	101	4.8%

* BPL Below Poverty Line.

**Table 3 vaccines-10-00507-t003:** Detailed reasons for non-vaccination during each call.

Reasons	Initial Call (*N* = 716)	1st Follow-Up Call (*N* = 292)	2nd Follow-Up Call (*N* = 160)
	*n* (%)	*n* (%)	*n* (%)
** *Health-related* **	459 (64.1)	118 (40.4)	66 (41.3)
Underweight	211 (29.5)	81 (27.7)	45 (28.1)
Child ill and not brought to the facility	106 (14.8)	36 (12.3)	18 (11.3)
Doctors not recommended	62 (8.7)	1 (0.3)	3 (1.9)
Baby born prematurely	39 (5.4)	none	None
Child ill, brought but not given vaccines	34 (4.7)	none	None
Delay in previous vaccination	7 (1)	none	None
** *Probable COVID-19 pandemic effects* **	239 (33.4)	99 (33.9)	59 (36.9)
Lockdown	86 (12.0)	57 (19.5)	29 (18.1)
Transportation problem	60 (8.4)	none	None
Insufficient staff (ASHA/ANMs)	29 (4.1)	40 (13.7)	30 (18.8)
Fear of coronavirus affecting the child	24 (3.4)	none	None
Not knowing time/place for vaccination	20 (2.8)	none	None
Insufficient vaccines	11 (1.5)	2 (0.7)	2 (1.3)
ASHA/ANM having COVID-19	4 (0.6)	none	None
Long wait	3 (0.4)	none	None
Insufficient number of babies	2 (0.3)	none	None
** *Lack of awareness/support* **	120 (16.8)	63 (21.6)	27 (16.9)
Lack of awareness about vaccine	43 (6)	27 (9.2)	11 (6.9)
Community worker/ANM did not inform	42 (5.9)	36 (12.3)	None
Lack of awareness about schedule	25 (5.7)	none	15 (9.4)
Fear/rumours about side effects/no faith	10 (1.4)	none	None
** *Logistical Reasons* ** *****	32 (4.5)	none	None
** *Other Reasons* ** ** ^†^ **	6 (0.8)	12	8 (5.0)

* mother too busy, family problems, out of the station, and mismatch of vaccination day and when they planned a visit. ^†^ baby crying, rain, festivals, and other personal reasons.

## Data Availability

The data presented in this study are available on request from the corresponding author. The data are not publicly available due to participant’s privacy.

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
