# Peer review of "Supporting Immunization Uptake during a Pandemic, Using Remote Phone Call Intervention among Babies Discharged from a Special Neonatal Care Unit (SNCU) in South India"

_vaccines, 2022, doi:10.3390/vaccines10040507_

Round 1
Reviewer 1 Report
The paper “Supporting immunization uptake during a pandemic, using remote phone call intervention among babies discharged from a Special Neonatal Care Unit (SNCU) in South India” aims at assessing the impact of phone-call counselling, on children immunization uptake during the COVID-19 pandemic in South India, to investigate factors (demographical and clinical) influencing the low vaccines uptake and increase the immunization uptake. This is a well-design and well-conducted study. Results of this study can be pivotal in further public health activities related to vaccination in India, having increase the vaccination uptake from 65.2% to 88.2%. It’s very touching read that most of non-vaccination were health-related including underweight and child illness.
Minor comments
- Remove the subtitle “CHALLENGES FOR IMMUNIZATION PROGRAMS IN INDIA”
- Reduce the introduction section moving sentences in page 2 lines 35-45 and page 3 lines 47-55 in the discussion
Author Response
-Remove the subtitle “CHALLENGES FOR IMMUNIZATION PROGRAMS IN INDIA”
Yes, we have fixed it in the revised version
-Reduce the introduction section moving sentences on page 2 lines 35-45 and page 3 lines 47-55 in the discussion
AS suggested, the introduction section has been revised
Reviewer 2 Report
This paper discuss Supporting immunization uptake during a pandemic, using remote phone call intervention among babies discharged from a 3 Special Neonatal Care Unit (SNCU) in South India. It is an interesting manuscript. However have some considerable limitations.
1- Selection Bias: authors should acknowledge that their sample size might be biased given the fact that the babies and families were all recruited form SNCU. The authors should discuss it further.
2- Self reporting bias is another major hindering and challenging limitation. The authors should dwell more on measure to decrease this bias and prevent it from affecting the results.
3- Introduction is very long and needs to be shortened and re-focused. Part of it should be moved ot the discussion.
4- The uathors need to add a section comparing phone use Vs other method described in the literature to increase and support immunization. A cost effectiveness model should be build
5- The manuscript should compared the results of other local Benschmarks, locally within India, and neighboring countries.
Author Response
1- Selection Bias: authors should acknowledge that their sample size might be biased given the fact that the babies and families were all recruited from SNCU. The authors should discuss it further.
We acknowledge it and have included it in the discussion section
2- Self-reporting bias is another major hindering and challenging limitation. The authors should dwell more on measures to decrease this bias and prevent it from affecting the results.
Thank you for your suggestion. We have included this limitation and have discussed it further.
3- The introduction is very long and needs to be shortened and re-focused. Part of it should be moved to the discussion.
We have made the necessary changes
4- The authors need to add a section comparing phone use Vs other methods described in the literature to increase and support immunization. A cost-effectiveness model should be build
Recommended information is added to the revised version
5- The manuscript should compare the results of other local Benchmarks, locally within India, and neighboring countries.
Recommended information is added to the revised version
Reviewer 3 Report
This is a much-needed report that corroborates common belief that an interest and a persistent effort such as two rounds of follow-up phone calls to the family can increase child immunization rate. The scheme is straightforward and the data supports authors' arguments. But the following points would improve the manuscript.
- Figures 1, 2, and 3 have titles only. Adding more detailed explanations of the figures in the legends could help the reader.
- In Table 1 and others, N and n should be explained. Also, changing the order of N and n in Table 1 would be better in reading the table.
- Some typos and sloppiness in the text can be seen in lines 60, 72, 104, 138 etc. Grammar and spell check is recommended.
Author Response
1. Figures 1, 2, and 3 have titles only. Adding more detailed explanations of the figures in the legends could help the reader.
Legends explaining the figure contents is added to each figure
2. In Table 1 and others, N and n should be explained. Also, changing the order of N and n in Table 1 would be better in reading the table.
We have added the explanation in the respective table.
3. Some typos and sloppiness in the text can be seen in lines 60, 72, 104, 138, etc. Grammar and spell-check are recommended.
Recommended changes are made in the revised version
Reviewer 4 Report
Supporting immunization uptake during a pandemic, using remote phone call intervention among babies discharged from a Special Neonatal Care Unit (SNCU) in South India.
Abstract
COVID-19 has impacted children’s immunization rates, putting the lives of children at risk. The present study assesses the impact of phone-call counseling, on immunization uptake during the pandemic. Families of babies discharged from the SNCUs in 6 government centers in 3 South Indian states were recruited. Calls were made 10 days after the immunization due date. Missed vaccinees were counseled and followed up on 7 and 15 days. Of 2313 contacted, 2097 completed the survey. Respondents were mostly mothers (88.2%), poor (67.5%), and had secondary level education (37.4% ). Vaccinations were missed due to the baby's poor health (64.1%), COVID-19 related concerns (32.6%), and lack of awareness (16.8%). At the end of the intervention, the immunization uptake increased from 65.2% to 88.2%. Phone-call intervention can safely support immunization and lower the burden on health workers.
Pls refer to this study which congruent with your study design
https://www.sciencedirect.com/science/article/abs/pii/S0011502902900313?casa_token=sQFRhjqsoIEAAAAA:wR5Do1lRckuFQhtv71n6JEKZEB6vKlUc9bN7XbEVxpdkxT6o8Uzkv_rcqIbvn2AUoJhlcq864dlm
We enrolled General Medical Service patients who received pharmacy-facilitated discharge from the hospital to home. The intervention consisted of a follow-up phone call by a pharmacist 2 days after discharge. During the phone call, pharmacists asked patients about their medications, including whether they obtained and understood how to take them. Two weeks after discharge, we mailed all patients a questionnaire to assess satisfaction with hospitalization and reviewed hospital records. Of the 1,958 patients discharged from the General Medical Service from August 1, 1998 to March 31, 1999, 221 patients consented to participate. We randomized 110 to the intervention group (phone call) and 111 to the control group (no phone call). Patients returned 145 (66%) surveys. More patients in the phone call than the no phone call group were satisfied with discharge medication instructions (86% vs. 61%, P = 0.007). The phone call allowed pharmacists to identify and resolve medication-related problems for 15 patients (19%). Twelve patients (15%) contacted by telephone reported new medical problems requiring referral to their inpatient team. Fewer patients from the phone call group returned to the emergency department within 30 days (10% phone call vs. 24% no phone call, P = 0.005). A follow-up phone call by a pharmacist involved in the hospital care of patients was associated with increased patient satisfaction, resolution of medication-related problems, and fewer return visits to the emergency department.
Take a particular note that one control group was included: We randomized 110 to the intervention group (phone call) and 111 to the control group (no phone call).
In order to make sure coherence of the study, whereby:
The primary goal of the present study was to examine the impact of a phone-based intervention on immunization uptake in SNCU babies. In addition, we also assessed:
- the extent of existing immunization coverage
- reasons for not vaccinating the child
- challenges faced by mothers during the postnatal period
The conclusion, therefore does not seem to reflect well the objective. This conclusion is over-claimed because a control is needed to confirm such claim:
Upsurge in immunization uptake in this study revealed that persistent reminder phone calls that provide information to families can be a viable, quick, safe, convenient, and a cost-effective alternative to connect with hard-to-reach populations and address their health-related needs in critical situations such as COVID-19 pandemic.
Other comments: the captions needs to be changed to reflect its contents
Figure 1. Follow-up Schedule.
Figure 2. Overall Reasons for Non-vaccination.
Table 1. Inclusion - exclusion criteria
Table 2. Sociodemographic characteristics (N=2097).
Table 3. Detailed Reasons for Non vaccination
Author Response
- The conclusion, therefore, does not seem to reflect well the objective. This conclusion is over-claimed because control is needed to confirm such a claim
Thank you for your suggestion. We have revised the conclusion section.
- Other comments: the captions need to be changed to reflect their contents
Figure 1. Follow-up Schedule.
Figure 2. Overall Reasons for Non-vaccination.
Table 1. Inclusion - exclusion criteria
Table 2. Sociodemographic characteristics (N=2097).
Table 3. Detailed Reasons for Non-vaccination
Title of all tables are further explained in detail
Reviewer 5 Report
On a first reading, this paper appears well-written and the study and the associated descriptive statistics analysis it reports are useful and interesing. I have no requirements for revision.
Author Response
Thank you for your feedback
Round 2
Reviewer 2 Report
Thank you for including all the suggested edits.
The paper is now ready for submission.